# Differences in Bacterial Communities of Retail Raw Pork in Different Market Types in Hangzhou, China

**DOI:** 10.3390/foods12183357

**Published:** 2023-09-07

**Authors:** Wen Wang, Zhengkai Yi, Wei Cai, Jiele Ma, Hua Yang, Min Zhou, Xingning Xiao

**Affiliations:** 1State Key Laboratory for Managing Biotic and Chemical Threats to the Quality and Safety of Agro-Products, MOA Laboratory of Quality & Safety Risk Assessment for Agro-Products (Hangzhou), Institute of Agro-Product Safety and Nutrition, Zhejiang Academy of Agricultural Sciences, Hangzhou, 310021, China; ww_hi1018@163.com (W.W.); zkyi121121@163.com (Z.Y.); jlma163@163.com (J.M.); yanghua@zaas.ac.cn (H.Y.); 2College of Food Science and Engineering, Wuhan Polytechnic University, Wuhan 430048, China; aieicw@163.com

**Keywords:** pork, retail markets, high-throughput sequencing, bacterial community, bacterial contamination

## Abstract

Pork is widely consumed globally, and pigs’ microbiota can potentially harbor foodborne pathogens. Contaminated pork in retail markets poses significant implications for food quality and safety. However, limited studies have compared pork microbiomes in various marketing environments. In this study, we utilized traditional microbial culture methods and high-throughput 16S rRNA sequencing to assess pathogen contamination and bacterial diversity in raw pork samples purchased from farmers’ markets and two types of supermarkets (upscale and ordinary) in Hangzhou, China. Traditional microbial plate cultures identified *E. coli* and *Salmonella* spp. in 32.1% (27/84) and 15.5% (13/84) of the collected pork samples, respectively. Moreover, 12 out of 13 *Salmonella* strains were found in farmers’ markets. The MIC results indicated a high prevalence of MDR strains, accounting for 51.9% in *E. coli* and 53.8% in *Salmonella*. The prevalence of NaClO tolerant strains was 33.3% and 92.3% for *E. coli* and *Salmonella*, respectively. Sequencing results indicated significantly higher microbial diversity in farmers’ market samples compared to supermarket samples. Farmers’ market pork samples exhibited a greater abundance of *Acinetobacter*, while *Pseudomonas* and *Brochothrix* were predominant in supermarket samples. The total abundance of pathogenic and spoilage bacteria was also higher for the farmers’ market samples. Cross-contamination during market trading was evident through a high correlation between bacterial abundance in pork from different stalls within the same farmers’ market. PICRUSt2 analysis identified significant differences in the average proportions of genes for carbohydrate, energy, and lipid metabolism from the farmers’ markets, suggesting an exacerbation of microbial metabolic activity and increased perishability of pork in this environment. In conclusion, this study revealed variations in the characteristics of raw pork bacterial contamination across different types of retail stores, as well as differences in the composition and diversity of their respective bacterial communities.

## 1. Introduction

China holds the global lead in pig breeding and pork production, with pork being a staple in its diet [1]. However, the large-scale processing, transportation, and storage of pork present significant opportunities for contamination by pathogenic and spoilage organisms, making it a major meat safety concern [2]. Studies have reported significant contamination levels of *Listeria monocytogenes*, *Salmonella*, and *Escherichia coli* in pork at retail markets, reaching up to 29%, 19.57%, and 30%, respectively [3,4]. Moreover, *Pseudomonas* spp. and *Brochothrix thermosphacta* have been found at initial concentrations of 5.9 to 6.4 log CFU/g in retail pork samples [5]. The presence of these microorganisms accelerates meat spoilage, leading to economic losses and food waste and compromising the microbiological safety of pork products, potentially causing foodborne diseases [6,7]. As reported, foodborne pathogens are responsible for more than 200 diseases, including typhoid fever, diarrhea, and cancer, and can lead to the death of unsuspecting consumers in both developing and developed countries [8]. In China, microbial pathogens remained the top cause of outbreak-associated illnesses, accounting for 41.7% of illnesses in 2020 [9].

To address such contamination issues, one preventive approach involves administering high levels of antimicrobial agents to farm animals through feed [10]. Moreover, disinfectants like sodium hypochlorite are commonly used for surface sterilization in food production facilities. However, the excessive and improper use of these sterilization methods has led to widespread bacterial tolerance, reducing their bactericidal effectiveness. This misuse has also resulted in significant residual antibacterial agents in both the environment and the food itself [11,12,13]. For example, a higher level of chlorine tolerance in antibiotic-resistant *Escherichia coli* was found when compared to antibiotic-susceptible strains [14]. Wu et al. (2015) reported that tolerance to antimicrobial agents at MIC > 512 mg/L was detected in 94.55% of isolates [15].

In China, pork is commonly sold in supermarkets and farmers’ markets, with the latter being traditional agricultural markets. At farmers’ markets, pig carcasses are displayed at room temperature for sale, while in supermarkets, the meat is always refrigerated, and strict safe handling procedures are followed, including transportation and storage through the cold chain mode. In contrast, farmers’ markets prioritize meat freshness and typically do not employ the cold chain mode, selling the meat on the same day it is slaughtered. The different marketing environments may expose the food to distinct environmental factors, potentially altering the native microbial composition or introducing non-native microorganisms as contaminants and promoting pathogen growth. Supermarkets invest significant resources in adhering to best practices in food handling, storage, and vending, ensuring proper hygiene through employee training [16]. Conversely, farmers’ markets often lack the necessary infrastructure for strict hygiene management and proper food storage. The open and confined nature of the food display stalls at farmers’ markets facilitates interactions with neighboring stalls, potentially promoting pathogen spread [17]. Bacterial contamination, inadequate personal hygiene, and unhygienic behavior can serve as channels for pathogen transmission to humans in these settings [18]. Additionally, consumer preferences for high-quality and fresh products have led to the popularity of upscale supermarkets, which are smaller in scale and focus on imported, fresh, and organic goods. With such a diverse array of pork retail facilities, a comprehensive assessment of contamination sources is imperative to ensure the safety of raw pork.

Traditional culture and molecular-based subtyping methods have long been utilized to investigate the prevalence of foodborne pathogens and cross-contamination pathways in markets [19,20]. However, these methods are laborious and time-consuming [21]. To address this, high-throughput sequencing technologies targeting the hypervariable region of 16S rRNA have emerged, enabling microbiome profiling through massively parallel sequencing. This approach significantly reduces analysis costs and time while enhancing the resolution of microbial community analysis [22]. In previous studies, this technology has been successfully applied to monitor the dynamic changes in microbial communities on pork surfaces [23]. However, the bacterial community compositions in raw pork sold in different markets remain unexplored.

The purpose of this study was to provide a comprehensive overview of the bacterial communities present in pork products at farmers’ markets and ordinary and upscale supermarkets. Specifically, we conducted a comparative analysis of raw pork samples obtained from these three market types in Hangzhou, China, using both traditional culture-dependent methods and the culture-independent approach of 16S rRNA amplicon sequencing. Our investigation into bacterial diversity and composition contributes to a deeper understanding of the bacterial communities present in raw pork.

## 2. Materials and Methods

### 2.1. Sample Collection

We collected a total of 28 sets of raw pork samples from 5 farmers’ markets and 10 supermarkets at 7 a.m. in Hangzhou, China. Each set included 250 g samples of tenderloin, hind leg meat, and pork belly in triplicate for a total of 84 samples. The sampling dates are 20 April and 6 May 2022. Farmers’ market samples were obtained from fresh pig carcasses from 3 randomly selected stalls, and a set of pork was taken from each stall for a total of 15 sets. The supermarket samples were grouped according to the pork brand and included 5 upscale supermarkets (USs) and 5 ordinary supermarkets (OSs). Supermarket samples were randomly sampled by pork brand. The information on collected samples is listed in Table 1. These samples were all packaged in polyethylene film, placed in an incubator with ice packs, and transported to the laboratory for preparation within 1 h.

### 2.2. Culture-Dependent Bacterial Isolation and Identification

Twenty-five grams of pork sample was placed in a sterile bag containing 100 mL of sterile saline solution and shaken for 2 min. The resulting solution (25 mL) was incubated in buffered peptone water medium (225 mL) at 37 °C with shaking at 100 rpm for 24 h [7]. Samples were then plated on MacConkey and Eosin Methylene Blue agar plates for isolation of *E. coli* and *Salmonella* spp., respectively. Molecular confirmation of *E. coli* isolates was performed using PCR targeting the *pho*A gene as previously described [24]. *Salmonella* isolates were identified by adding 1 mL of suspension to 10 mL of tetrathionate broth and incubating at 42 °C with shaking at 100 rpm for 24 h. Culture samples were then plated on Xylose Lysine Tergitol 4 agar and further confirmed on Chromogenic *Salmonella* agar. Molecular confirmation of presumptive isolates was carried out using PCR amplification of the *inv*A gene, as previously described [25].

### 2.3. Antimicrobial Susceptibility Testing

The minimum inhibitory concentrations (MIC) of *Salmonella* and *E. coli* isolates utilized the broth microdilution method as prescribed by the Clinical and Laboratory Standards Institute (CLSI) [26]. The 0.5 McFarland inoculum suspensions were further diluted at 1:100 in Mueller–Hinton broth, resulting in an inoculum density of 5 Log CFU/mL. The antibiotic panel was reconstituted by introducing 200 µL of the inoculum into each well, followed by incubation at a temperature of 37 °C for 18 h. The 13 antimicrobials examined included aztreonam (resistant breakpoint, ATM ≥ 16 μg/mL), cefotaxime sodium (CEF ≥ 8 mg/mL), cefotaxime (CTX ≥ 4 µg/mL), meropenem (MEM ≥ 4 µg/mL), gentamicin (GEN ≥ 16 µg/mL), amikacin (AMK ≥ 64 µg/mL), ciprofloxacin (CIP ≥ 1 µg/mL), tigecycline (TIG ≥ 8 µg/mL), tetracycline (TET ≥ 16 µg/mL), sulfamethoxazole (T/S ≥ 4/76 µg/mL), colistin (CS ≥ 8 µg/mL), florfenicol (FFC ≥ 16 µg/mL), and fosfomycin (resistant, FOS ≥ 256 μg/mL). *Salmonella* and *E. coli* strains resistant to at least three classes of antimicrobials were defined as multidrug-resistant (MDR). *Salmonella enteritidis* CVCC 1806 and *E. coli* ATCC 25922 were used as the quality control organisms.

### 2.4. NaClO Tolerance Determinations

The NaClO tolerance determination method utilized in this study was referred to in our previous study [27]. In brief, the MIC of sodium hypochlorite (NaClO) was determined by the broth microdilution method. Bacterial cultures of 0.5 McFarland units were diluted 1:100 with Mueller–Hinton broth. A 56.8 mg/mL NaClO stock solution was made and confirmed using a ChlorSense meter (Palintest, Erlanger, KY, USA), and dilutions were aliquoted to the diluted bacterial cultures. These MIC assays were performed in 96-well microtiter test plates. Each well contained 100 µL of NaClO solution and was then inoculated with 100 µL of suspended bacterial cultures to a final inoculum density of 5 Log CFU/mL per well. *S. enteritidis* CVCC 1806 and *E. coli* ATCC 25922 were used as the quality control organisms. Preliminary tests indicated MIC values of 128 mg/L for both organisms, and tolerance was defined as >128 mg/L NaClO.

### 2.5. DNA Extraction

All collected pork samples (*n* = 84) were used for 16S rRNA amplicon sequencing. The DNA extraction method utilized in this study was based on the approach described by Niamah et al. (2012) with slight modifications [28]. The mixture of 30 mL of sterile saline and pork sample obtained in part 2.2 was centrifuged at 5000 rpm for 5 min, and the supernatant was transferred to a new centrifuge tube and centrifuged at 12,000 rpm for 5 min. The precipitate was used for bacterial genomic DNA isolation using a TIANamp fecal DNA kit (Tiangen Biotechnology, Beijing, China) according to the manufacturer’s instructions. DNA concentrations were measured using a Nanodrop One spectrophotometer (Thermo Fisher, Pittsburg, PA, USA), and the DNA quality was confirmed by 1% agarose gel electrophoresis. Each sample was stored at −20 °C until PCR analysis.

### 2.6. PCR Amplification and Sequencing

The V3-V4 region of the bacterial small-subunit 16S rRNA gene was amplified with primers 341F (5′-CCTACGGGNGGCWGCAG-3′) and 805R (5′-GACTACHVGGGTATCTAATCC-3′). Validation of PCR amplification product was performed according to Qiu et al. [7]. The amplified products were sequenced on an Illumina NovaSeq platform at LC-Bio Technology Co. Ltd. (Hangzhou, China).

### 2.7. Sequence Data Analysis

Paired-end reads were assigned to samples based on their unique barcode and truncated by removing the barcode and primer sequences, and the resulting sequences were then merged using FLASH (v1.2.11) to an average length of 426 bp [29]. Sequencing quality was assessed using Fastqc. Quality filtering on the raw reads was performed under specific filtering conditions to obtain high-quality clean tags using fqtrim (v0.9.4). Chimeric sequences were filtered using Vsearch software (Version 2.3.4). Amplicon sequence variants (ASVs) were obtained after dereplication using DADA2 [30].

Alpha and beta diversity scores were calculated by normalization to the same sequences randomly. Alpha diversity is applied for analysis of the complexity of species diversity for a sample with five indices: Chao1, Observed species, Goods coverage, Shannon, and Simpson, and were calculated using QIIME2 [31]. Beta diversity refers to species differences between different environmental communities, and we utilized principal coordinate analysis (PCoA) and clustering analysis (UPGMA) that were calculated with QIIME2. Graphs were drawn using the R package (v3.5.2). The taxonomy of each ASV was analyzed by SILVA (release 138) classifier. The PICRUSt2 program (v2.2.0-b) based on the Kyoto Encyclopedia of Genes and Genomes (KEGG) database was used to predict functional pathways used by the microbiota in different samples. The representative ASV in FASTA format and a Biological Observation Matrix (BIOM) table of the abundance of each ASV across each sample were used as inputs for PICRUSt2. The BIOM table was generated using Python package biom (v.2.1.7). PICRUSt2 was also used to calculate the nearest sequenced taxon index as a measure of prediction uncertainty for the microbiota data sets between two groups [32].

### 2.8. Statistical Analysis

SPSS 20.0 software package (IBM, Chicago, IL, USA) was used for statistical analysis. The Kruskal–Wallis test was used for diversity differential analysis. ANOSIM was performed based on the Bray–Curtis dissimilarity distance matrices to identify differences in microbial communities between different groups. The data were shown as mean ± standard deviation, and the significance level was set at *p* < 0.05.

## 3. Results and Discussion

### 3.1. Prevalence, Antibiotic and Chlorine Resistance of Foodborne Pathogens

A total of 84 raw pork samples were collected from retail markets and screened for pathogenic bacteria. Among these samples, *Salmonella* was isolated from 13 samples (15.5%), and *E. coli* was detected in 27 samples (32.1%). The distribution of isolates in the supermarkets (SMs) and farmers’ markets (FMs) is shown in Table 2. Notably, 12 out of 13 *Salmonella* isolates were from FMs, while 12 and 15 *E. coli* isolates were from FMs and SMs, respectively. The MIC results indicated a high prevalence of MDR strains, accounting for 51.9% in *E. coli*., and 53.8% in *Salmonella*. Additionally, the prevalence of NaClO tolerance strains was 33.3% and 92.3% for *E. coli* and *Salmonella* spp., respectively. These findings reveal an alarmingly high level of drug resistance and chlorine tolerance in pork sold in the market, pointing to the serious issue of antibiotic and disinfectant abuse in the process of pig breeding in China. Previous studies have also demonstrated a high prevalence of foodborne pathogens in chicken and pork slaughterhouses, as well as their downstream retail markets in China, indicating a potential transmission risk along the entire slaughterhouse and production chain to the retail market [3,33]. The elevated levels of detection, MDR, and NaClO tolerance phenotypes of these pathogenic bacteria are concerning, as meat can be contaminated at various stages along the slaughter and distribution pipeline [34,35]. Moreover, this is likely a significant factor contributing to the frequent occurrence of food infections caused by pathogenic bacteria in meat samples. Hence, the detection of pathogenic bacteria in retail meat is an indispensable part of food microbiological analysis, allowing for early identification and mitigation of potential health risks to consumers.

### 3.2. Bacterial Community Richness and Diversity

The 16S rRNA sequencing data of the samples were used to determine the *α* diversity of the bacterial populations based on the ASV differences (Appendix A). The Chao1 and Shannon indices indicate the number and diversity of species [36]. The Chao1 and Shannon indices for the farmers’ market samples were significantly higher than those of both types of supermarkets, indicating a more complex structure and higher diversity of bacterial communities in pork from FM samples (Figure 1). Additionally, the FD sample had a significantly lower Chao1 index score than the other groups and possessed the lowest microbial richness across the FM samples. The two types of supermarkets did not display any significant differences in the Shannon index (*p* > 0.05), but the Chao1 index for the ordinary supermarket (OS) samples was significantly lower than that of the upscale supermarket (US) samples (*p* < 0.05), indicating that the microbial diversity in pork from the two supermarkets was consistent although the US samples had higher microbial abundance. Based on Chao1 and Shannon indices, raw pork in OS samples had lower richness and diversity in their bacterial communities.

Furthermore, we performed a beta diversity analysis of bacterial community composition using weighted UniFrac distances and applied PCoA to compare differences between the FM and SM samples. The beta diversity scores showed partial overlap, suggesting a similarity in bacterial compositions between the two types of samples. However, within the ordinary supermarket (OS) samples, the composition structure exhibited more differences compared to the FM and upscale market (US) samples (Figure 1C). Additionally, we conducted a comparison based on the pork sample type and generated confidence circle plots, which nearly overlapped, indicating a lack of significant differences in the bacterial community structures (Figure 1D).

### 3.3. Comparison of Bacterial Communities of Different Types of Pork Markets

The results of 16S rRNA gene sequencing revealed a diverse microbial community in all samples, comprising 60 phyla, 152 classes, 342 orders, 582 families, 1354 genera, and 2421 species. At the phylum level, the dominant phyla were Proteobacteria, Firmicutes, Bacteroidetes, and Actinobacteriota, which together accounted for an average of 98.79% of the total ASVs (Figure 2A), consistent with the previous study [37]. At the genus level, the most abundant genera were *Acinetobacter*, *Brochothrix*, *Pseudomonas*, *Photobacterium*, *Psychrobacter*, *Aeromonas*, and *Weissella* (Figure 2B). These genera are commonly associated with food spoilage [38,39,40]. *Acinetobacter*, in particular, was the most prevalent genus across all samples and is known to thrive in thermophilic conditions, making it well suited to the temperatures typically found in slaughterhouses [41]. It is also commonly found in raw meat [42], making open-air farmers’ markets the most favorable environment for its growth. On the other hand, SM samples exhibited a higher abundance of *Brochothrix* and *Pseudomonas*, which are psychrophiles responsible for shelf life reductions in fresh food stored at low temperatures. Among them, *Brochothrix thermosphacta* is one of the most abundant spoilage organisms in fresh and cured meats and fish products due to its ability to tolerate high salt and low pH levels and grow at refrigeration temperatures [43]. Its presence causes spoilage characterized by discoloration, gas production, and a pungent cheesy odor [44]. Previous studies on bacterial communities in pork have reported *Acinetobacter* as the primary contaminant in pork stored at room temperature, while *Brochothrix* and *Pseudomonas* were the dominant genera in pork stored at refrigerated temperatures of 4 °C [45]. These findings are in line with the results of the current study. Interestingly, our study revealed high levels of *Photobacterium*, a genus not previously reported in studies of raw pork, but commonly found in deep ocean environments associated with fish spoilage in seafood [46]. The presence of *Photobacterium* in frozen pork samples suggests its potential role in spoilage [37]. It is worth noting that Photobacterium is sensitive to heat and can only grow at temperatures below 25 °C, requiring a minimum level of Na^+^ for sustained growth [47]. The lack of previous detection of *Photobacterium* in raw pork microbiota composition studies can be attributed to the culture media and temperatures commonly used to cultivate meat spoilage bacteria, which do not support the growth of *Photobacterium*. Consequently, this genus might have been overlooked in previous studies. In our particular FM samples, the FD samples displayed a low abundance of *Acinetobacter* and a high abundance of *Photobacterium*, suggesting that the sample was not freshly slaughtered meat and may have been refrigerated. This disparity in storage environment possibly accounts for the FD sample having the lowest alpha diversity scores among the five FM samples, highlighting the significant impact of temperature on the composition of the microbial community.

To further investigate the disparities and resemblances between FM and SM fresh pork, we employed clustering based on Bray–Curtis distances. At the phylum level, we did not observe any noticeable separation of samples, and only some SM samples formed distinct clusters. Conversely, when we analyzed at the genus level, there was clear clustering based on location. FM samples displayed distinct characteristics compared to SM samples, whereas no evident differentiation was observed between OS and US samples. These results indicated that the microbial community composition in raw pork is closely linked to factors such as storage temperature and environment rather than the size of the supermarket.

At the species level, our analysis revealed the presence of some typical pathogenic and spoilage bacteria in the samples. The pathogenic bacteria include the groups *Escherichia_Shigella* unclassified, *Staphylococcus* unclassified, *Salmonella enterica*, *Proteus mirabilis*, and *Streptococcus pneumoniae*. These pathogens are associated with various illnesses, such as urinary tract infections, bacteremia, pneumonia, diarrhea, septicemia, and meningitis. Notably, the relative abundance of these pathogenic bacteria was found to be higher in FM samples compared to SM samples (Figure 3). A comparison was conducted between two identification techniques, revealing that the culture-dependent approach exhibited concordance with the 16S rRNA findings in detecting *Salmonella*. However, the culture-based approach failed to detect *E. coli* in FA, USD, and OSB, whereas the results from HTS indicated its presence in all samples (Figure 3 and Table 2). For spoilage bacteria, FM samples exhibited a higher presence of *Weissella* spp., with *Weissella viridans* being the most prevalent species in the meat processing industry. This bacterium is known for its rapid growth and ability to produce green mucus on the surface of pork, resulting in the pork becoming sticky and appearing green [48]. *Pseudomonas fragi* is a bacterium known for causing meat spoilage under aerobic storage conditions. Microbial contamination in meats can be attributed to various factors, such as unsanitary preparation places, inadequate clean utensils, cross-contamination from raw meat, poor personal hygiene, and the unhygienic practices of meat handlers and vendors [49]. Consequently, potential contamination of the meat from the stalls is high for FM due to a lack of basic clean infrastructure and services and poses a risk of salmonellosis, listeriosis, typhoid fever, and diarrhea [50].

Traditionally, the characterization of microbiota in food matrices is performed by standard cultivation and phenotyping methods. However, the ‘gold standard’ plate count method underestimates the diversity of complex bacterial communities, as <1% of environmental bacteria are not culturable using routine methods [51]. The implementation of culture-independent approaches has delivered substantial insights into microbial community profiling, complementing traditional methods. A disadvantage, however, is that a reliable identification can only go down to the genus level, and no distinction can be made between viable and non-viable bacteria [52]. Thus, as described in the human gut microbiota [53], culture-dependent and independent approaches can complement each other in the investigation of microbial populations in foods.

### 3.4. Bacterial Cross-Contamination in the Pork Market

Spearman correlation analysis was applied to investigate correlations between bacterial contamination in the FM and SM locations (Figure 4). Interestingly, a strong correlation was observed between samples from three different stalls within the same FM. Notably, the correlation was found to be the lowest in FD, the market that possessed the lowest alpha diversity scores (Figure 4A, Appendix A). This indicated the possibility of cross-contamination between stalls in the other four FM locations, contributing to higher bacterial community diversity. Furthermore, the correlation coefficients were higher in the US samples compared to the OS, which was consistent with the alpha diversity levels (Figure 4B, Appendix A).

### 3.5. Microbial Function Prediction

We also performed a functional profiling of samples using PICRUSt and the KEGG pathway database (Figure 5). ATP-binding cassette (ABC) transporter was the most abundant pathway found across all groups. These transporters exist widely in bacteria and utilize the energy generated by ATP hydrolysis to expel substances out of the cell across a concentration gradient and are prominent in conferring multidrug resistance [54], which may explain the abundance of MDR *E. coli* and *Salmonella* identified in our study. We also observed significant differences in multiple pathways of amino acid, carbohydrate, and energy metabolism between the FM and SM groups. For instance, the citric cycle, an aerobic catabolic pathway, was more represented in the FM groups. There were also significant differences in glyoxylate and dicarboxylate metabolism, as well as arginine and proline metabolism, which are related to the formation of biogenic and volatile amines [37]. The abundance of genes related to fatty acid metabolism and oxidative phosphorylation was also higher in FM samples (Figure 5). These differences in the average proportions of carbohydrate, energy, and lipid metabolic pathways indicated that the FM environment enhances the metabolic activities of microorganisms, making pork more susceptible to spoilage.

## 4. Conclusions

In this study, we revealed differences in the composition and diversity of the bacterial communities in raw pork from farmers’ markets and supermarkets. The normal temperature and open-air sales environment of the farmers’ market contributed to a more complex and diverse bacterial community structure, as well as increased bacterial cross-contamination. This study highlights the importance of combining high-throughput sequencing with traditional plate culture methods to assess overall bacterial communities and identify potential pathogens. The findings may have implications for public health, particularly for consumers purchasing pork from different retail markets, and it is necessary to adopt proper follow-up handling and consumption methods to ensure food safety.

## Figures and Tables

**Figure 1 foods-12-03357-f001:**
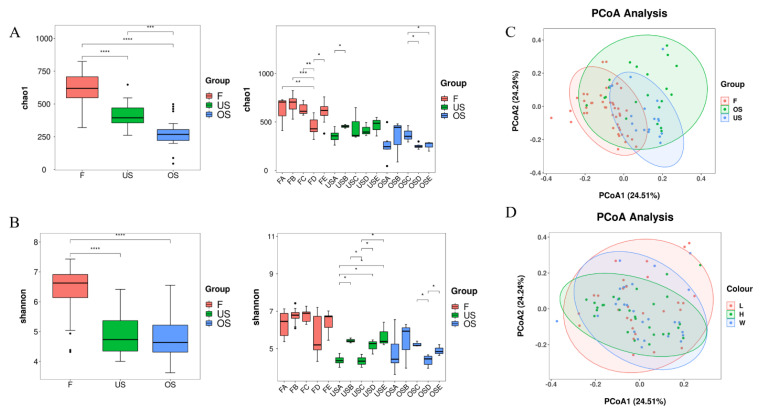
Diversity of microbial populations identified from 84 pork samples used for this study. Alpha diversity according to sampling sites for (**A**) Chao1 and (**B**) Shannon indices. PCoA of bacterial communities by (**C**) sampling site and (**D**) meat sample location. F, farmers’ market; US, upscale market; and OS, ordinary supermarket. Asterisks indicate statistical significance by F test (* *p* < 0.05; ** *p* < 0.01; *** *p* < 0.001; **** *p* < 0.0001).

**Figure 2 foods-12-03357-f002:**
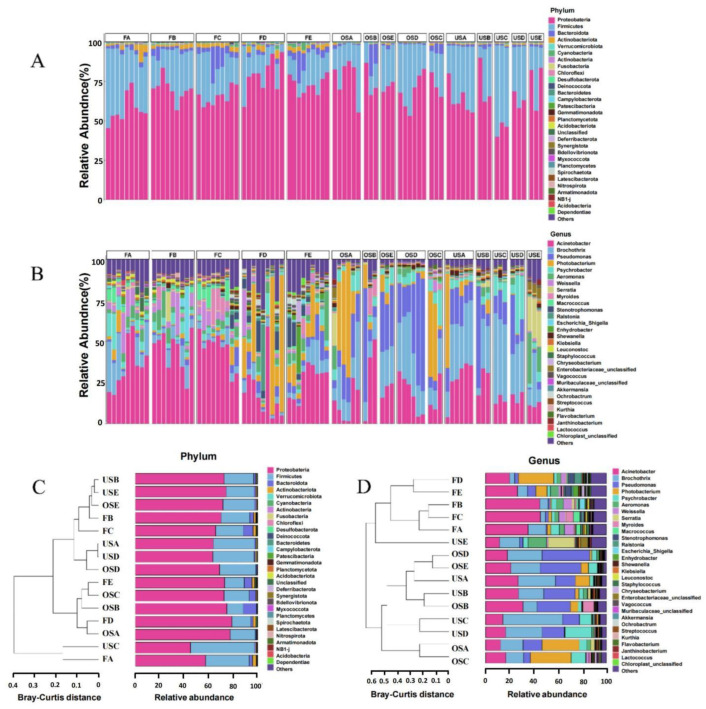
Changes in relative abundance (%) of the microbial community in pork samples at the phylum (**A**) and genus (**B**) level. Cluster analysis in the pork samples at the phylum (**C**) and genus (**D**) level.

**Figure 3 foods-12-03357-f003:**
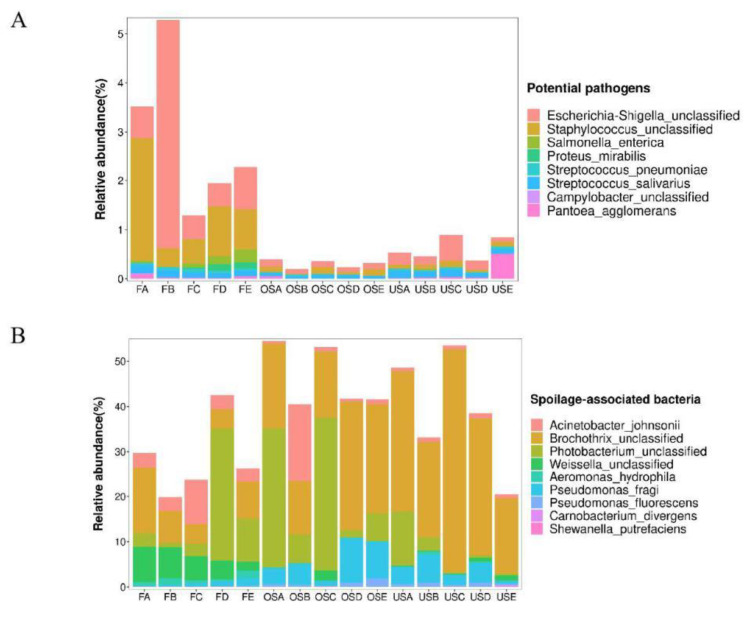
Relative abundance of pathogenic (**A**) and spoilage (**B**) bacteria.

**Figure 4 foods-12-03357-f004:**
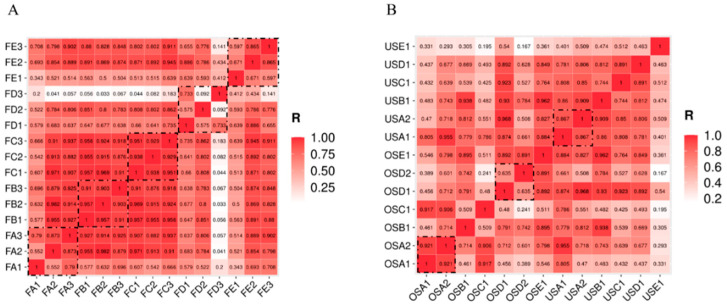
Correlation analysis of bacterial taxons over all samples. (**A**) Farmers’ markets and (**B**) supermarkets. The dashed box is the correalation with different stalls in the same samplesites.

**Figure 5 foods-12-03357-f005:**
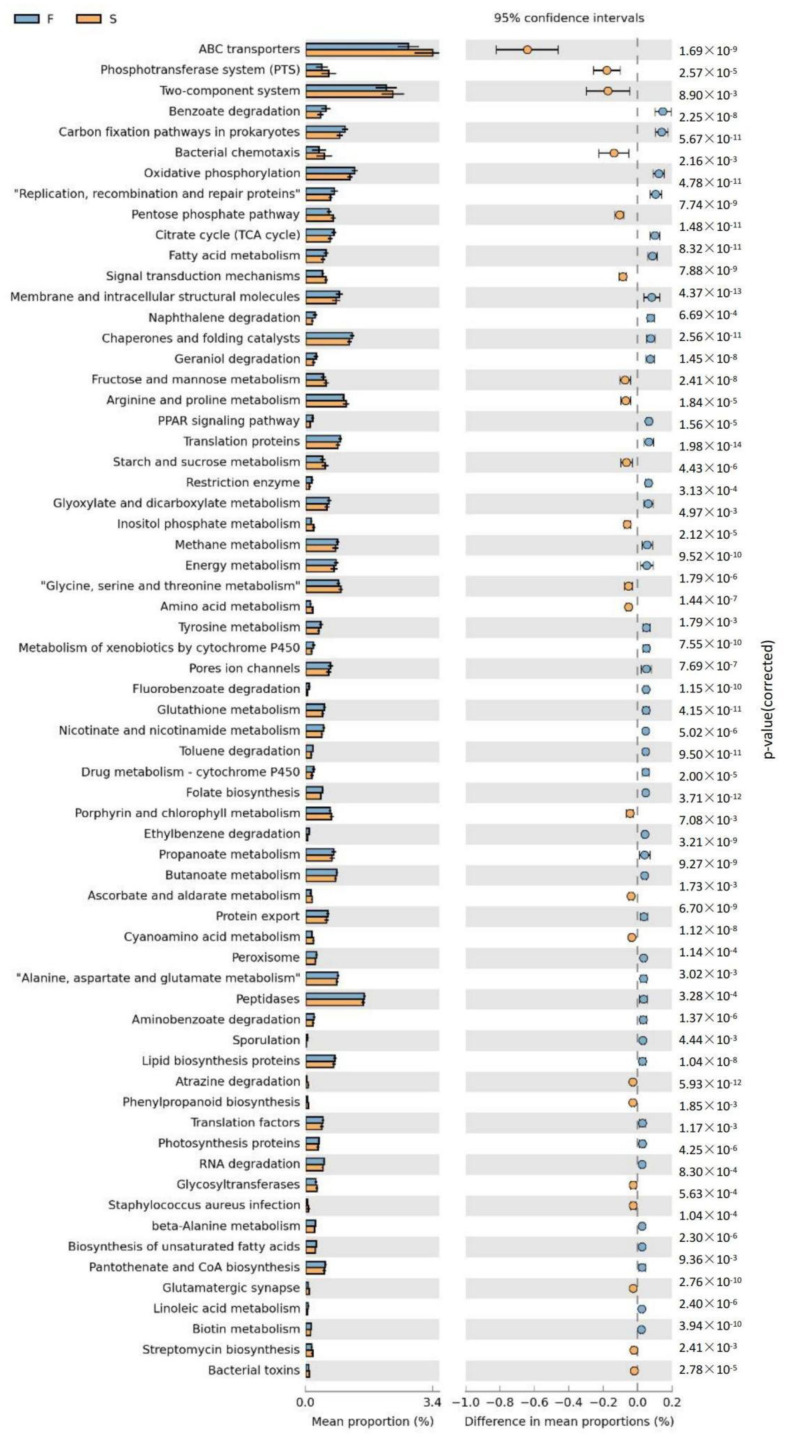
Predictive functional profiling. 16S rRNA gene sequences were analyzed using PICRUSt.

**Table 1 foods-12-03357-t001:** Detailed information on collected pork samples.

Supermarket Type	Sampling Sites	Sample Code	Sample Size
Farmers’ Market	Farmers’ Market A	FA1	3
		FA2	3
		FA3	3
	Farmers’ Market B	FB1	3
		FB2	3
		FB3	3
	Farmers’ Market C	FC1	3
		FC2	3
		FC3	3
	Farmers’ Market D	FD1	3
		FD2	3
		FD3	3
	Farmers’ Market E	FE1	3
		FE2	3
		FE3	3
Upscale Supermarket	Upscale Supermarket A	USA1	3
		USA2	3
	Upscale Supermarket B	USB	3
	Upscale Supermarket C	USC	3
	Upscale Supermarket D	USD	3
	Upscale Supermarket E	USE	3
Ordinary Supermarket	Ordinary Supermarket A	OSA1	3
		OSA2	3
	Ordinary Supermarket B	OSB	3
	Ordinary Supermarket C	OSC	3
	Ordinary Supermarket D	OSD1	3
		OSD2	3
	Ordinary Supermarket E	OSE	3

**Table 2 foods-12-03357-t002:** Prevalence, multidrug resistance, and NaClO resistance of *E. coli* and *Salmonella* in samples from different markets.

	Sample Code	Sample Size	*E. coli*			*Salmonella*		
			Prevalence (%) *	MDR (%) ^#^	NaClO Tolerance (%) ^#^	Prevalence (%) *	MDR (%) ^#^	NaClO Tolerance (%) ^#^
Farmers’ Market	FA	9	0 (0%)	-	-	1 (11.1%)	0	-
	FB	9	2 (22.2%)	0	-	1 (11.1%)	1 (100%)	1 (100%)
	FC	9	4 (44.4%)	0	-	4 (44.4%)	2 (100%)	4 (100%)
	FD	9	1 (11.1%)	1 (100%)	1 (100%)	3 (30%)	2 (66.7%)	3 (100%)
	FE	3	2 (22.2%)	2 (100%)	1 (50%)	3 (30%)	1 (33.3%)	3 (100%)
Upscale Supermarket	USA	6	3 (50%)	2 (66.7%)	2 (66.7%)	0 (0%)	-	-
	USB	3	2 (66.7%)	0	-	0 (0%)	-	-
	USC	3	2 (66.7%)	1 (50%)	-	0 (0%)	-	-
	USD	3	0 (0%)	-	-	0 (0%)	-	-
	USE	3	1 (33.3%)	1 (100%)		0 (0%)	-	-
Ordinary Supermarket	OSA	6	2 (33.3%)	1 (50%)	-	1 (16.7%)	1 (100%)	1 (100%)
	OSB	3	0 (0%)	-	-	0 (0%)	-	-
	OSC	3	1 (33.3%)	0	1 (33.3%)	0 (0%)	-	-
	OSD	6	6 (100%)	6 (100%)	3 (50%)	0 (0%)	-	-
	OSE	3	1 (33.3%)	0	1 (100%)	0 (0%)	-	-

* The value in parentheses is the ratio of positive samples to the total number of samples. ^#^ The value in parentheses is the percentage of multidrug-resistant/NaClO-resistant isolates (% of total isolates).

## Data Availability

The 16S rDNA sequencing data were submitted to the NCBI Sequence Read Archive (SRA) database, accession no. PRJNA914695.

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
