# Peer review of "Differences in Bacterial Communities of Retail Raw Pork in Different Market Types in Hangzhou, China"

_foods, 2023, doi:10.3390/foods12183357_

Round 1
Author Response
The review is well conceptualized and presents very useful data for an overview of the differences in the composition and diversity of the bacterial communities in raw pork from the farmers markets and supermarkets in Hangzhou, China.
It is systematically written, and compared pork microbiomes in various marketing environments.
The topic is relevant, and these findings may have implications for public health, particularly for consumers purchasing pork from a different retail market. It is also address to adopt proper follow-up handling and consumption methods to ensure food safety.
The conclusions are in accordance with the presented arguments, and they do address the main questions posed. The references aren’t all appropriate, and tables and figures are clearly and obviously displayed.
Although, there are some minor revisions before publication.
Specific comments:
- There are a some of references errors in the text, many of which are inappropriate.
For example:
Line 39/ref. 1
Line 45 /ref. 5,6
Line 49/ref. 8,9
Line 310/in Table 3…some of references
..etc.
I will not list them all, but it takes a lot of concentration and corrections to put everything back in order so that there is an accurate citation in the text.
Reply: The references have been thoroughly checked and revised throughout the document to ensure that they are accurately cited within the text.
- Line 109 and 131/there is no reference for the method.
Reply: The references Qiu et al. (2022) and Xiao et al. (2022) have been added in the method as suggested (lines 120 and 146)
Qiu, M., et al., Dynamic Changes of Bacterial Communities and Microbial Association Networks in Ready-to-Eat Chicken Meat during Storage. Foods, 2022. 11(22).
Xiao, X., et al., Chlorine Tolerance and Cross-Resistance to Antibiotics in Poultry-Associated Salmonella Isolates in China. Front Microbiol, 2021. 12: p. 833743.

Reviewer 2 Report
Dear author(s), dear editor(s),
The manuscript is well written and the experiment is well controlled. The study contributes to the knowledge in the field.
Some revisions to manuscript are needed:
Table 2. It is confusing. Please make it clearer. E.g. two contaminating samples is related to FB2 or? And 4 for FC2 which is not possible as n=3? And give explanations for the numbers in parentheses. Maybe summarized results could be presented for replicates as for HTS data (e.g. Figure 3)?
Table 3. It is too banal (too general). Please remove it or incorporate with Figure 3. Or give the information about the samples and abundance in Table 3 and you can skip Figure 3.
Figure 2. Difficult to read. The font is too small.
L 139 Please indicated how many samples were used for HTS e.g n=84
L290 “…and found that the culture-dependent approach was consistent with 16S rRNA results for the detection of Salmonella but not E. coli” please explain in which way? Was E.coli found in all samples or only in those not detected by the culture-based approach?
Overall, the manuscript points out biases due to culture-dependent methods. Please provide information on the biases in HTS.
Author Response
The manuscript is well written and the experiment is well controlled. The study contributes to the knowledge in the field.
Some revisions to manuscript are needed:
Item 1: Table 2. It is confusing. Please make it clearer. E.g. two contaminating samples is related to FB2 or? And 4 for FC2 which is not possible as n=3? And give explanations for the numbers in parentheses. Maybe summarized results could be presented for replicates as for HTS data (e.g. Figure 3)?
Reply: As suggested, the samples were summarized to make the results clearer in the revised Table 2, and the explanatory notes for the numbers in parentheses were added in the form of footnotes (line 225).
Item 2: Table 3. It is too banal (too general). Please remove it or incorporate with Figure 3. Or give the information about the samples and abundance in Table 3 and you can skip Figure 3.
Reply: Table 3 has been removed as suggested.
Item 3: Figure 2. Difficult to read. The font is too small.
Reply: An enhanced version of Figure 2 has been incorporated into the text (line 339).
Item 4: L 139 Please indicated how many samples were used for HTS e.g n=84
Reply: All collected pork samples (n=84) were used for 16S rRNA amplicon sequencing. This has been added in Line 157.
Item 5: L290 “…and found that the culture-dependent approach was consistent with 16S rRNA results for the detection of Salmonella but not E. coli” please explain in which way? Was E.coli found in all samples or only in those not detected by the culture-based approach?
Reply: E.coli was found in all samples in the results of HTS in Figure 3, while it has not been detected in FA, USD, and OSB by the culture-based approach. The sentence has been revised as “A comparison was conducted between two identification techniques, revealing that the culture-dependent approach exhibited concordance with the 16S rRNA findings in detecting Salmonella. However, the culture-based approach failed to detect E. coli in FA, USD, and OSB, whereas the results from HTS indicated its presence in all samples” to make it clear (Lines 311-315).
Item 6: Overall, the manuscript points out biases due to culture-dependent methods. Please provide information on the biases in HTS.
Reply: A disadvantage of HTS is that a reliable identification can only go down to genus level, and no distinction can be made between viable and non-viable bacteria. The information has been added in Lines 333-335.

Reviewer 3 Report
Dear Editors and authors,
1-The abstract of the manuscript needs to add some results to be more clear and comprehensive for the reader.
2-The introduction needs to add a paragraph about food contamination with pathogenic bacteria to be more expanded.
3- 2.3. Antimicrobial Susceptibility Testing, The authors did not mention the size of the inoculum? How many microorganisms are there in this volume? What is the culture medium used and what is the incubation temperature?
This method is opaque.
4-. NaClO Tolerance Determinations, In this method also, the authors did not mention the bacterial size used? The authors did not cite a source for this method . I suggest you to read (Xiao, X., Bai, L., Wang, S., Liu, L., Qu, X., Zhang, J., ... & Wang, W. (2022). Chlorine tolerance and cross-resistance to antibiotics in poultry-associated Salmonella isolates in China. Frontiers in Microbiology, 12, 833743.)
5-. DNA Extraction, This method needs a scientific reference, I suggest you, (Niamah, A. K. (2012). Detected of Aero gene in Aeromonas hydrophila isolates from shrimp and peeled shrimp samples in local markets. Journal of Microbiology, Biotechnology and Food Sciences, 2(2), 634-639. )
6-Line 190 , E. coli spp., correct .
7- The conclusions contain some results that should be removed.
Quality of English Language is good.
Author Response
Dear Editors and authors,
- The abstract of the manuscript needs to add some results to be more clear and comprehensive for the reader.
Reply: As suggested, more detail results on “antimicrobial susceptibility” and “NaClO Tolerance” of bacterial isolates have been added in the abstract (lines 19-21).
- The introduction needs to add a paragraph about food contamination with pathogenic bacteria to be more expanded.
Reply: Pork contamination with pathogenic and spoilage bacteria has been illustrated in Lines 41-52, and the foodborne disease has been added in Lines 52-56. In addition, we added the chlorine tolerance in bacteria to expand the introduction in Lines 63-66.
- 3. Antimicrobial Susceptibility Testing, The authors did not mention the size of the inoculum? How many microorganisms are there in this volume? What is the culture medium used and what is the incubation temperature? This method is opaque.
Reply: The 0.5 McFarland inoculum suspensions were further diluted at 1:100 in Mueller-Hinton broth, resulting in an inoculum density of 5 Log CFU/mL. The antibiotic panel was reconstituted by introducing 200 µL of the inoculum into each well, followed by incubation at a temperature of 37â—¦C for 18 hours. It has been revised in the manuscript (Lines 132-135).
- NaClO Tolerance Determinations, In this method also, the authors did not mention the bacterial size used? The authors did not cite a source for this method. I suggest you to read (Xiao, X., Bai, L., Wang, S., Liu, L., Qu, X., Zhang, J., ... & Wang, W. (2022). Chlorine tolerance and cross-resistance to antibiotics in poultry-associated Salmonella isolates in China. Frontiers in Microbiology, 12, 833743.)
Reply: As suggested, further information regarding the methodology employed for the determination of NaClO tolerance has been incorporated into the text. Furthermore, we have included our earlier study (Xiao et al., 2022) as a citation in this work. (Lines 145-153)
- DNA Extraction, This method needs a scientific reference, I suggest you, (Niamah, A. K. (2012). Detected of Aero gene in Aeromonas hydrophila isolates from shrimp and peeled shrimp samples in local marketJournal of Microbiology, Biotechnology and Food Sciences, 2(2), 634-639.)
Reply: As suggested, the reference for the DNA Extraction method has been added and this has been illustrated in lines 158-159.
- Line 190, coli spp., correct.
Reply: It has been corrected as “E. coli” in Line 210.
- The conclusions contain some results that should be removed.
Reply: As suggested, the description of results in conclusions have been deleted (lines 383-391).
